# In Tandem Control of La-Doping and CuO-Heterojunction on SrTiO_3_ Perovskite by Double-Nozzle Flame Spray Pyrolysis: Selective H_2_ vs. CH_4_ Photocatalytic Production from H_2_O/CH_3_OH

**DOI:** 10.3390/nano13030482

**Published:** 2023-01-25

**Authors:** Pavlos Psathas, Areti Zindrou, Christina Papachristodoulou, Nikos Boukos, Yiannis Deligiannakis

**Affiliations:** 1Department of Physics, University of Ioannina, 45110 Ioannina, Greece; 2Institute of Nanoscience and Nanotechnology (INN), NCSR Demokritos, 15310 Athens, Greece

**Keywords:** strontium titanate, doping, La:SrTiO_3_, CuO, heterojunction, FSP, double-nozzle, photocatalysis, hydrogen, CH_4_ selectivity

## Abstract

ABO_3_ perovskites offer versatile photoactive nano-templates that can be optimized towards specific technologies, either by means of doping or via heterojunction engineering. SrTiO_3_ is a well-studied perovskite photocatalyst, with a highly reducing conduction-band edge. Herein we present a Double-Nozzle Flame Spray Pyrolysis (DN-FSP) technology for the synthesis of high crystallinity SrTiO_3_ nanoparticles with controlled La-doping in tandem with SrTiO_3_/CuO-heterojunction formation. So-produced La:SrTiO_3_/CuO nanocatalysts were optimized for photocatalysis of H_2_O/CH_3_OH mixtures by varying the La-doping level in the range from 0.25 to 0.9%. We find that, in absence of CuO, the 0.9La:SrTiO_3_ material achieved maximal efficient photocatalytic H_2_ production, i.e., 12 mmol g^−1^ h^−1^. Introduction of CuO on La:SrTiO_3_ enhanced selective production of methane CH_4_. The optimized 0.25La:SrTiO_3_/0.5%CuO catalyst achieved photocatalytic CH_4_ production of 1.5 mmol g^−1^ h^−1^. Based on XRD, XRF, XPS, BET, and UV-Vis/DRS data, we discuss the photophysical basis of these trends and attribute them to the effect of La atoms in the SrTiO_3_ lattice regarding the H_2_-production, plus the effect of interfacial CuO on the promotion of CH_4_ production. Technology-wise this work is among the first to exemplify the potential of DN-FSP for scalable production of complex nanomaterials such as La:SrTiO_3_/CuO with a diligent control of doping and heterojunction in a single-step synthesis.

## 1. Introduction

Photocatalytic storage of light photons in the form of fuels such as H_2_ or CH_4_ is globally envisaged as an environmentally benign technology [1,2]. To this front, in the last decades, photocatalytic semiconductors receive great consideration [3].

Strontium Titanate (SrTiO_3_) has a classical ABO_3_ perovskite structure, with the ideal cubic lattice [4]. SrTiO_3_ has received enormous attention as a photocatalyst, oxidative catalyst, or catalyst-support, due to beneficial characteristics such as low price [5], chemical stability, and excellent thermal stability, e.g., melting point as high as 2080 °C, with carbon and sulfur tolerance, adaptability and modifiable oxidative properties [6]. SrTiO_3_ has a highly reducing conduction-band-edge energy position (E_CB_) of −1.2 eV vs. NHE [7,8], which makes it a highly efficient photocatalyst for hydrogen production from H_2_O [9]. SrTiO_3_ has the disadvantage of a broad 3.2 eV band gap, permitting only the absorption of UV photons, thus to remediate this drawback, a common technique is to apply dopants or heterojunction with other materials [10].

So far, synthesis of nanocrystalline SrTiO_3_ has been achieved with various methods, each having distinct advantages/disadvantages [4,11], e.g., solid phase [12] that is a low-cost approach, however, produces rather large particles, sol-gel [11] that gives pure phase small nanoparticles at a higher cost, hydrothermal [5] that produces controllable size but requires long synthesis-times with impurity possibility. Moreover, doping of SrTiO_3_ with an appropriate dopant-cation at low concentration, and the formation of heterojunctions between SrTiO_3_ and selected co-catalytic particles, pose additional hurdles, complexity, and cost of the synthesis process [13]. In this context, it is of great importance to develop synthesis technologies for high-quality SrTiO_3_ in scale-up production, at the same time allowing doping of SrTiO_3_ and heterojunction SrTiO_3_/MO_x_ formation in one step.

The Flame Spray Pyrolysis (FSP) process is an aerosol combustion technology for nanoparticle synthesis [14,15,16] that is well-established for industrial-scale production [17,18]. As such, FSP is already used in industry to produce several commonly used metal oxides [19]. FSP utilizes high-temperature combustion, i.e., 1000–2000 K [20], of a precursor containing metal atoms dispersed in an organic solvent, that is sprayed in the form of aerosol droplets. Appropriate selection of precursor mixtures and process parameters allows precise control of the crystal phases [19], devoid of chemical leftovers. So far, FSP has been established for the successful production of a wide range of single-metal oxides [19], however, establishing FSP process parameters for the production of ABO_3_ perovskite nanomaterials is typically more challenging, e.g., such as avoiding the formation of the separate oxides AO_n_ and BO_m_. The relative importance of process parameters has been exemplified in our recent works on FSP-engineering of NaTaO_3_ [21], Bi_2_Fe_4_O_9_ and BiFeO_3_ [22,23], W-, Zr-Doped BiVO_4_ [24], and by Kudo and Amal for BiVO_4_ [25], as well as Abe and Laine for La_2_Ti_2_O_7_ [26]. Very recently, the synthesis of SrTiO_3_ [27,28] by FSP has been achieved and used for catalytic combustion of CO and CH_4_, however with no reference to photocatalytic evaluation or optimization.

Lanthanum (La) doping of SrTiO_3_ has been shown to increase the electrochemical efficiency of the perovskite [29,30], and as co-dopant with other atoms [31,32]. Domen [33,34] and Kudo [35,36] have demonstrated that SrTiO_3_ co-doped with La and Rh exhibits high solar-to-hydrogen energy conversion efficiency for water splitting to H_2_. Moreover, selected cations, e.g., La^3+^, Ce^3+^, or Nitrogen [8,37] allow control of structural/photo/electronic properties of doped-SrTiO_3_, enhancing the photocatalytic efficiency [38,39]. Thus, SrTiO_3_ offers a versatile template, that can be optimized towards specific technologies either via doping or heterojunction engineering.

Surficial heterojunctions of SrTiO_3_ with pertinent metal-oxides allow control of the selectivity of surface reactions. More specifically, Cu-atoms and their particles (CuO, Cu_2_O, Cu^0^) are particularly attractive in photocatalytic processes, since they have been shown to control selectivity towards specific products, i.e., Cu_2_O exhibits selectivity towards CH_3_OH production and Cu^0^ towards hydrocarbons and C_2_ products [40,41]. In particular, studies show that CuO clusters on the TiO_2_ surface can serve as an efficient co-catalyst to enhance H_2_ production [42,43]. Meanwhile, Cu_2_O on TiO_2_ was also reported to be more suitable for H_2_ production, which was attributed to the better energy band alignment between TiO_2_ and Cu_2_O, which facilitates charge separation [44,45].

Herein, we employ a Single-Nozzle FSP (SN-FSP) process for the synthesis of La:SrTiO_3_ and a Double-Nozzle FSP (DN-FSP) creating a heterojunction of CuO finely dispersed on La:SrTiO_3_ in one-step [46,47], see Figure 1. Recently, our group utilized DN-FSP to produce TiO_2_ loaded with noble metal particles (>2 nm), to significantly enhance photocatalytic hydrogen production [48]. In the present work, our hypothesis was that an in tandem La-doping of SrTiO_3_ heterojunctions with CuO can allow control of the SrTiO_3_ electronic/photocatalytic properties, as well as controlled selectivity in photocatalytic products. In this context, our specific aims were: (i) To develop an FSP protocol for the synthesis of La:SrTiO_3_ with controlled La-doping; (ii) To advance the FSP protocol to allow heterojunction of CuO on La:SrTiO_3_; (iii) To study the H_2_ vs. CH_4_ photocatalytic process in comparison to CuO. For this reason, we have performed photocatalytic studies in an H_2_O/CH_3_OH mixture. CH_3_OH is a well-known hole-scavenger [49], however, most publications do not examine the CH_3_OH involvement in the reaction path and the final products, e.g., eventually CH_4_.

## 2. Materials and Methods

### 2.1. Flame Spray Pyrolysis (FSP) Synthesis of SrTiO_3_ Nanoparticles

The precursor solution contains Strontium acetate (STREM), Titanium(VI) isopropoxide (97%, Aldrich) for the synthesis of perovskite SrTiO_3_. For the deposition of cocatalytic metals, i.e., La atoms, with Lanthanum Acetylacetonate (97%, STREM). The Sr and Ti precursors were dispersed in a mixture of Acetic acid and Xylene (1:1 volume ratio), while for the La precursor a mixture of toluene and 2-EHA (1:1 volume ratio) that consisted of approximately 10% of the total volume of the final precursor solution. We underline that it is the control of FSP parameters that allows Sr- and Ti- to be engaged exclusively in the formation of the SrTiO_3_ crystals, while La- is introduced as a lattice-dopant.

*Double-Nozzle FSP*: In the DN-FSP as shown in Figure 1, two FSP-nozzles operate in tandem that are asymmetrically positioned, so that the system generates two different kinds of nanomaterials by controlling the properties of each nozzle independently, where the left FSP-nozzle contained the Sr/Ti/La precursors, while the right FSP-nozzle contained Cu(NO_3_)_2_·3H_2_O (Supelco) in Acetonitrile:Ethyleglygol (1:1 volume ratio). Screening experiments were conducted to find the preferent geometric parameters: α_1_ nozzle angle at 20^o^, and α_2_ = 20^o^. The internozzle distance was placed at x = 8 cm and the vertical intersection distance of the two flames, above the nozzle, was b = 10, see Figure 1C.

**Figure 1 nanomaterials-13-00482-f001:**
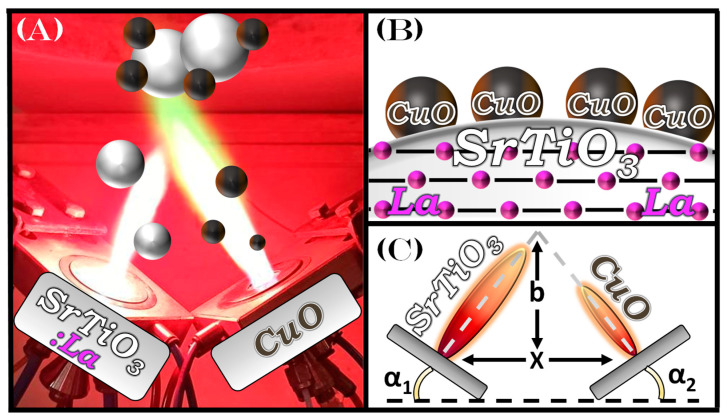
(**A**) Description of the Double-Nozzle FSP (DN-FSP) set-up, with the SrTiO_3_ formation in the left-side nozzle and the CuO formation in the right-side nozzle. (**B**) Schematic depiction of the structure of the La:SrTiO_3_/CuO particle. (**C**) The geometric parameters for the two nozzles in DN-FSP.

The FSP parameters for the synthesis of the nanoparticles consisted of an oxygen dispersion flow rate of D = 5 L min^−1^ (Linde 99.999%) and a precursor flow rate of P = 5 mL min^−1^. These D and P rates were also used for both nozzles in the DN-FSP setup. The pilot flame was ignited by premixed O_2_ and CH_4_ (4 L min^−1^, 2 L min^−1^). In the Single-Nozzle as well as in the Double-Nozzle FSP, with the assistance of a vacuum pump (BUSCH), the produced particles were deposited on a glass microfiber filter with a binder (Albet Labscience GF_6_257) and collected by scrubbing the nanoparticles from the filter. The nanomaterials were collected in glass vials under an inert Argon atmosphere, until use.

For convenience, herein the produced materials, listed in Table 1, are codenamed as follows: X_La:STO/Y_Cu where STO = SrTiO_3_, X the nominal % La-content per weight of SrTiO_3_, Y = the nominal % Cu-content per weight of SrTiO_3_. In this way, we have prepared 0.9La:STO and 0.35La:STO, respectively, listed in Table 1. For the DN-FSP experiments, we have produced STO/2Cu, STO/1.2Cu, and STO/0.5Cu, respectively. Finally the 0.25La:STO/0.5Cu produced by DN-FSP is codenamed as La:STO/Cu.

### 2.2. Photocatalytic Evaluation

Photocatalytic experiments were performed in a double-walled photochemical reactor (Toption instrument co. Ltd.) with a total reaction volume of 340 mL, at a temperature of 25 ± 2 °C, controlled by a recirculation chiller cooling system. The UV source was a 250 W Mercury lamp, positioned at the geometrical center of the photoreactor inside the quartz-immersion well. The irradiation power at the experimental mean distance of 3 cm was 0.5 W cm^−2^ as measured with a power meter (Newport model, 1918-C). 

A Gas-Chromatography System combined with a Thermal-Conductivity-Detector (TCD-Shimadzu GC-2014, Carboxen 1000 column, Ar carrier gas) was used to identify and quantify the produced H_2_ and CH_4_ gases. In each experiment, 50 mg of the catalyst was suspended in 200 mL Milli-Q water and 50 mL methanol (20% per volume) as a hole-scavenger. Photo-deposition of Pt-cocatalyst was implemented to increase the photocatalytic production, using hydrogen hexachloroplatinate (IV) hydrate, (H_2_Pt_4_Cl_6_·H_2_O, 99.9%, Alfa Aesar). The error bars of approximately 8%, for the photocatalytic products H_2_ and CH_4_, reflect the detection uncertainty and statistical deviation after three catalytic runs.

### 2.3. Characterization of Materials

XRD measurements were carried out to identify the crystal phase identification and structural properties of the materials, using a Bruker D8-Advance diffractometer with a Cu source (Kα, λ = 1,5418 Å), with operation parameters of 40 KV generator voltage and 40 mA current. The particle crystal size (*d*_XRD_) as obtained from the XRD data was calculated with the Scherrer Equation (1)
(1)$$d_{\mathrm{XRD}}=\frac{K\;\lambda }{\left(\mathrm{FWHM}\right)\times \;\cos\theta }$$
where K = 0.9, λ = 1,5418 Å, and FWHM is the full-width at half-maximum of the XRD peaks [50].

Transmission Electron Microscopy analysis was used to map the particle morphology. The amount of La-atoms and Cu-atoms of the nanomaterials was calculated by a home-built Energy-Dispersive X-Ray Fluorescence (EDXRF), and an annular 241 Am radioisotopic source was used for sample excitation. The source is fixed coaxially above a CANBERRA SL80175 Si(Li) detector (5 mm crystal thickness, 80 mm^2^ area), with a 25 μm thick Be window and an energy resolution of 171 eV for the 5.9 keV Mn Kα line. Spectral analysis was carried out using the WinQxas software package (International Atomic Energy Agency, 1997–2002). Quantitative analysis was based on in-house standard samples and the construction of calibration curves.

The morphology and phase composition of the La:STO/Cu particle was studied using an FEI Talos F200i field-emission (scanning) transmission electron microscope (Thermo Fisher Scientific Inc., Waltham, MA, USA) operating at 200 kV, equipped with a windowless energy-dispersive spectroscopy microanalyzer (6T/100 Bruker, Hamburg, Germany).

The Specific Surface Area (SSA) and the pore size of the nanomaterials were measured by a Quantachrome NOVAtouch_LX2 to record the N_2_ adsorption-desorption isotherms at 77 K. The SSA was calculated using the absorption data points in the range of 0.1−0.3 relative pressure P/Po. While the pore radius analysis was obtained by the BJH method [51] in the range of 0.35–0.99 P/Po. 

Diffuse-Reflectance UV-Vis absorption spectra were recorded with a Perkin Elmer Lambda-35 spectrometer with BaSO_4_ powder used as a background standard, operating at room temperature for the wavelength range of 200–800 nm, while the band gap energy (E_g_) was calculated using the Kubelka–Munk method [52]. 

The oxidation state of the Sr, Ti, O, and Cu atoms was monitored by X-ray photoelectron spectroscopy (XPS), using a SPECS spectrometer equipped with a twin Al-Mg anode X-ray source and a multi-channel hemispherical sector electron analyzer (HSA-Phoibos 100), a monochromatized Mg Kα line at 1253.6eV, analyzer pass-energy at 15 eV, and the base pressure at 2–5 × 10^−9^ mbar. The binding energies were determined vs. the energy of C1s carbon peak at 284.5 eV. The peak deconvolution was performed employing mixed Gaussian–Lorentzian functions, using WinSpec software, developed at the Laboratoire Interdisciplinaire de Spectroscopie Electronique, University of Namur, Belgium.

## 3. Results

### 3.1. SrTiO_3_-Based Perovskite Nanoparticle Synthesis by FSP

Highly crystalline SrTiO_3_ has been successfully produced by Single-Nozzle FSP, see XRD data in Figure 2A. The distinct peaks 2θ at 22.71, 32.34, 39.91, 46.43, 52.29, 57.74, 67.78, and 77.13 degrees, confirm the formation of pure cubic perovskite SrTiO_3_ structure (JCPDS74–1296). All diffraction peaks are assigned to Miller indices of the (100), (110), (111), (200), (210), (211), (220), and (310) planes for the cubic perovskite structure (Pm3m), with no traces of other crystalline byproducts, such as TiO_2_ or Sr-oxide. 

The *d*_XRD_ values calculated by the Scherrer method [50], listed in Table 1, indicate *d*_XRD_ sizes in the range of 45–55nm. The TEM images of the pristine SrTiO_3_, show the formation of quasi-spherical particles with a distribution of particle sizes, see Figure 2B, as commonly observed in FSP-made particles [19]. Specifically, according to TEM, the FSP-made SrTiO_3_ includes a few large particles with diameters of 40–50nm and many smaller particles with diameters below 20nm. The ensuing particle-size distribution calculated from the TEM images, Figure 2C, shows a mean size of *d*_TEM_ = 17 ± 0.2nm as obtained from a Gaussian fitting. Comparison of *d*_XRD_ and *d*_TEM_ exemplifies the well-known effect of large particles to predominate the diffraction peaks in XRD, thus *d*_XRD_ overestimates the average particle size.

**Table 1 nanomaterials-13-00482-t001:** Structural Characteristics of La:SrTiO_3_, SrTiO_3_/CuO, and La:SrTiO_3_/CuO nanoparticles.

Nanomaterial ^1^	La-, Cu-Content XRF Analysis (%wt)	*d*_XRD_ (nm) (±4)	SSA (m^2^ g^−1^) (± 0.5)	Total Pore Volume (cm^3^ g^−1^) (±0.02 × 10^−2^)	Band Gap E_g_ (eV) (± 0.1)
* Single-Nozzle FSP *					
** *Pristine SrTiO_3_* **	La:0 /Cu:0	45	32.3	0.14 × 10^−2^	3.2
** *0.9La:STO* **	La:0.93 ± 0.05 /Cu:0	47	53.1	0.39 × 10^−2^	3.2
** *0.3* ** ** *5* ** ** *La:STO* **	La:0.36 ± 0.05 /Cu:0	55	57.5	0.36 × 10^−2^	3.2
* Double-Nozzle FSP *					
** *STO/2Cu* **	La:0.05 ± 0.05 /Cu:2 ± 0.1	53	34.9	0.12 × 10^−2^	3.2
** *STO/1.2Cu* **	La:0/Cu:1.2 ± 0.1	49	32.1	0.11 × 10^−2^	3.2
** *STO/0.5Cu* **	La: 0.05 ± 0.05 /Cu:0.5 ± 0.1	45	32.0	0.11 × 10^−2^	3.2
** *La:STO/Cu* **	La: 0.26 ± 0.05 /Cu:0.5 ± 0.1	52	37.3	0.16 × 10^−2^	3.2

^1^ X_La:STO/Y_Cu where STO = SrTiO_3_, X = the nominal % La-content per weight of SrTiO_3_, Y = the nominal % Cu-content per weight of SrTiO_3_.

In Figure 3A, HRTEM images for La:STO/Cu are shown. Distinct Miller planes of CuO (110) are resolved with d = 2.75 Å [53], showing that CuO particles are deposited on the surface of the SrTiO_3_ particle. These CuO particles are small, <2 nm, therefore they are not detected in the XRD. Figure 3C–F present the scanning-TEM images for the Ti, Sr, La, and Cu atoms mapping for material La:STO/Cu. Importantly, Figure 3E demonstrates that the La atoms are evenly dispersed throughout the whole volume of the SrTiO_3_ particle. This verifies the atomic distribution of La, i.e., without La-clusters. The STEM for Cu atoms, Figure 3D, verifies the formation of dense/particle structures on SrTiO_3_. Caution is drawn to the fact that the faint blue hue in Figure 3D is the enhanced emissions from Cu-grid atoms employed for the TEM measurements that are in proximity to the Sr-atoms. These are secondary Cu-electrons enhanced via Strontium excitations. Thus, in Figure 3F, only the particles inside the white circle are actual CuO particles, showing that CuO particles are found only on the surface of the SrTiO_3_. 

La-doped SrTiO_3_ produced by Single-Nozzle FSP retain their high crystallinity and purity, i.e., no secondary phases are formed, such as La_2_O_3_. La-doping is evidenced by a pink-hue color developed in La:SrTiO_3_ vs. the white SrTiO_3_. Upon increasing La-doping, the SrTiO_3_-particle size tends to increase, i.e., from 45 nm *d*_XRD_ to 55 nm *d*_XRD_. Strikingly, the Specific Surface Area increases also, see Table 1. This counterintuitive observation is further analyzed in the BET-data analysis hereafter.

Double-Nozzle FSP used to form the SrTiO_3_/CuO heterojunctions, retains the high crystallinity of SrTiO_3_, see XRD data in Figure 2A, with a tendency towards larger particle sizes at increased Cu-content. In agreement with previous reports [28], this can be attributed to the increased contribution to the increased enthalpy that is added due to the second flame, thus increasing the overall synthesis temperature, and extending the temperature at the stage of agglomeration. 

The N_2_-adsorption isotherms for selected nanomaterials are presented in Figure 3A, while the data for nanomaterials with different CuO percentages are shown in Appendix A in the Supporting Information. All nanomaterials have the characteristic of a type-IV isotherm. The SSA values and pore-volume analysis, see Figure 4B, reveal some interesting trends: in the Single-Nozzle FSP, the La-dopped particles show an increase in their SSA values, see Table 1. However, taking into account the XRD data, this increased SSA is not concurring with the particle size. More insightful information is obtained by examining the pore size and pore volume trends, see Figure 4B, which reveals an increase in pore volume upon La-doping. Typically, in literature, SrTiO_3_ particles are reported to possess a total pore volume in the range of 15 cm^3^ g^−1^ [18,54], which is in the same range observed for our pristine SrTiO_3_ nanomaterial, Appendix A in the Supporting Information. Upon La-doping, a sharp increase is observed for the SSA, but more importantly a 3-fold increase in the pore volume to 39 cm^3^ g^−1^. This trend can be attributed to a geometrical effect as depicted in the scheme in Figure 4. This is a result of the FSP process, where the La-doping decreases the packing/aggregation of the SrTiO_3_, even though some SrTiO_3_ might grow bigger. It is worth mentioning the relatively high surface area and pore volume in our nanomaterials, i.e., compared vs. previous synthesis methods for SrTiO_3_ that possessed low SSA, i.e., due to the application of high calcination temperatures [5,55]. 

### 3.2. Spectroscopic Characterization

#### 3.2.1. Diffuse Reflectance UV-Vis Spectroscopy

SrTiO_3,_ as a semiconductor, has been shown to possess an indirect bandgap of 3.25 eV and a direct bandgap of 3.75 eV [56]. It has been shown that La or Nb-dopings, or oxygen vacancies can produce an *n*-type alteration in the Density of States (DOS) [56], i.e., extra DOS are formed within the band gap right below the Conduction Band bottom. In our data in Figure 5B, the indirect band gap has been calculated using the Kubelka–Munk method [52]. Notice that its absorption starts at 390 nm [57], resulting in all the nanomaterials having Eg values close to 3.2 eV, listed in Table 1. Lanthanum/Copper incorporation in the SrTiO_3_ materials fundamentally changed the color of the nanoparticles, with a purple hue observed after the lanthanum doping, while a drastic brown tint was observed for the 2% CuO heterostructure [28], which is clearly visible from extra absorption in the 400–580 nm range, Figure 5A.

Overall, the present DRS-UV-Vis analysis verifies that the electronic structure of the SrTiO_3_ semiconductors produced by SN-FSP and DN-FSP, as well as their trends upon La-doping and Cu-heterojunction, are in accordance with literature data. 

#### 3.2.2. X-ray Photoelectron Spectroscopy

XPS data for Sr- or O-atoms are presented in Figure 6A,C, respectively. Figure 6B presents Cu-XPS data, there is an emphasis on copper in order to confirm the oxidation state of the Cu-NPs. The material STO/2Cu, i.e., with the higher Cu-loading is exemplified, since it possessed strong Cu-XPS signals, to ascertain the Cu-oxidation state with precision. La-could not be detected by XPS in any of our materials.

In Figure 6B, the binding energies located broadly at 932.8eV and 952.7eV correspond to the Cu 2p_3/2_ and Cu 2p_1/2_, respectively [58,59], while the presence of a strong satellite signal at 941eV and the broader peaks of Cu 2p_3/2_ and Cu 2p_1/2_ indicate that the Cu atoms are in the Cu(II) oxidation state [58,59]. Thus, XPS results confirm that the heterostructure is indeed SrTiO_3_/CuO. This is in agreement with the FSP settings, i.e., the P/D = 5/5 produces a typical oxidizing environment [19] that in our Cu-precursor promotes the formation of CuO. The binding energies at 132.2 eV and 134 eV were identified as the Sr 3d_5/2_ and Sr 3d_3/2_, corresponding to the Sr^2+^ state of the SrTiO_3_ [60,61], Figure 6A. For the oxygen species, Figure 6C, three peaks were observed, with the binding energies at 528.9 eV that can be attributed to lattice oxygen species of the SrTiO_3_ crystal structure [61,62]. The 531.1 eV is attributed to chemisorbed oxygen, and lastly, the peak at 532.4 eV is attributed to adsorbed oxygen on the particle surface or hydroxyl groups [61,62]. The oxygen species from the partially substituted materials, i.e., 0.25%La/0.5%CuO have distinctly different intensity ratios of the (lattice) vs. (chemisorbed) oxygen species. This can be possibly attributed to oxygen vacancies that are created from the bending of the lattice, impacting the adsorption and efficiency of the oxygen species at the catalyst surface [61,62]. For the Ti 2p, all materials have peaks located at 457.7 eV and 463.4 eV, which are the binding states of Ti 2p_3/2_ and Ti 2p_1/2_, respectively, that correspond to typical Ti^4+^ states, Appendix A in the Supporting Information. 

Overall, the present XPS data show that the FSP-made nano SrTiO_3_ consists of typical Ti^4+^ states with some O vacancies in all materials. In STO/Cu, the CuO particle was confirmed by XPS in accordance with the oxidizing FSP process used herein. 

### 3.3. Photocatalytic Evaluation

Photocatalytic results for photocatalytic hydrogen production of the materials were evaluated using a mixture of H_2_O/CH_3_OH as a catalytic substrate. The only products were H_2_ or CH_4_, in all cases. The gas-production data are shown in Figure 7A,B for H_2_ or CH_4_, respectively. The corresponding rates are presented in Figure 7C,D, respectively. The highest H_2-_yield was 11980 umol g^−1^ h^−1^, achieved by the 0.9%La:SrTiO_3_, which is ~500% higher than for pristine SrTiO_3_ with H_2-_yield 2760 umol g^−1^ h^−1^. A clear beneficial trend is observed, i.e., higher La-doping promotes H_2_ production. This trend is in agreement with the literature [31,32]. Herein, however, we focus attention on the relative rates for CH_4_ also. 

From Figure 7 we observe that in the absence of CuO, the production of CH_4_ from the photoreduction of methanol was minimal. The SrTiO_3_/CuO heterostructure portrays very different results, with the 2%CuO having almost the same H_2_ production as the pristine SrTiO_3_, however, there is a sharp increase in the CH_4_ (76 to 136 umol g^−1^ h^−1^). This selectivity towards CH_4_ is apparent for all CuO heterojunctions. Decreasing CuO content resulted in higher H_2_ and CH_4_ production, 2%CuO increasing the production to 704 umol g^−1^ h^−1^ CH_4_. Notice that with increasing Cu-loading the SrTiO_3_ particle surface is fully covered with CuO, which in turn might act as an inhibitor of light-absorbance by the SrTiO_3_. This ‘darkening’ effect plays a key role in diminishing the photocatalytic activity, i.e., both H_2_ and CH_4_. Most interestingly, the material 0.25%La/0.5%Cu showed an enhanced H_2_/CH_4_ selectivity, with the CH_4_ reaching 1469 umol g^−1^ h^−1^ and H_2_ to 5907 umol g^−1^ h^−1^. 

Overall, the present data in Figure 7 show that: (i) in all instances, La-doping greatly increases the photocatalytic activity, (ii) CuO on SrTiO_3_ drives the products toward the reduction of CH_3_OH to CH_4_. Importantly, after the photocatalytic application, the particles fully retained their structure as evidenced by XRD data, see Appendix A in the Supporting Information.

## 4. Discussion

The present data show that FSP offers a versatile technology for the production of nano SrTiO_3_, La:SrTiO_3,_ SrTiO_3_/CuO, and La:SrTiO_3_/CuO. These materials show significant photocatalytic activity that can be tailored towards either pure H_2_ production or CH_4_/H_2_ production from an H_2_O/CH_3_OH mixture. Table 2 allows a comparison of H_2_ photogeneration by the present FSP particles vs. literature, SrTiO_3_-based nanocatalysts. From Table 2, and the literature generally, the use of methanol or ethanol as a sacrificial agent to enhance CH_4_ production has not been reported so far. Most research publications utilize Xenon radiation, with the highest yields shown in Table 2, although there are several with UV irradiation. 

In the case of SrTiO_3_, full exploitation of its band gap > 3.2 eV dictates the use of UV light, in order to maximize the H_2_ yield [63,64,65,66,67], in comparison with visible-light H_2_ production [68,69]. In this context, Table 2, shows that the FSP-made 0.9%La:SrTiO_3_ nanocatalyst has the highest yield vs. literature data on pertinent La:SrTiO_3_ materials, although the irradiation method that was used herein (UV-Lamp) definitely boosts the photocatalytic efficiency. So far, stabilizing CuO or Cu_2_O NPs on metal oxides with lower conduction-band positions is a common strategy to enhance photocatalytic H_2_ production compared with pristine CuO and Cu_2_O [70,71]. In this context, it was reported that coupling Cu_2_O with TiO_2_ or ZnO leads to higher H_2_ photoproduction [70,72]. Although there are no previous data on CH_4_ production promotion by SrTiO_3_/CuO or La:SrTiO_3_/CuO heterojunctions, some insights can be gathered from other systems. For example, Xiong et al. demonstrated that on a TiO_2_-Pt-Cu_2_O nanohybrid, Cu_2_O promotes the CH_4_ production and at the same time suppresses the H_2_ evolution, whereas Pt favors H_2_O activation [73]. Moreover, Chen et al. had shown that the co-existence of Cu^1+^/Cu^0^ species on TiO_2_ enhances the photocatalytic efficiency, by increasing the lifetime of electrons leading to an enhancement of CO_2_ hydrogenation. In the same study, the reduction of CuO to Cu^0^ was found to be more efficient for the photocatalytic production of CH_4_. A Cu-Cu_2_O/TiO_2_ hybrid has shown an excellent selectivity for CH_4_ which is attributed to the suppression of CO formation from Cu^0^ species while Cu^1+^ species act as the active sites for CH_4_ production [74]. All these data pose the possibility that the promotion of CH_4_ production by SrTiO_3_/CuO or La:SrTiO_3_/CuO heterojunctions might involve the formation of Cu^1+^ or Cu^0^ species at the SrTiO_3_/CuO interface. 

## 5. Conclusions

In the present work, we introduce Double-Nozzle Flame Spray Pyrolysis technology as a synthesis method for production of nano SrTiO_3_ perovskites, with in tandem control of La-Doping of SrTiO_3_ crystal plus a CuO/SrTiO_3_-heterojunction. The proposed FSP technology allows controlled production of La:SrTiO_3_, SrTiO_3_/CuO, or La:SrTiO_3_/CuO with fundamental advantages of the one-step production, and the potential for scalable production of complex nanomaterials.

The resulting nanomaterials showed distinct structural-electronic properties, with the La-doping inducing a characteristic increase in SSA via the formation of larger pore voids. Diligent control of the La-doping and SrTiO_3_/CuO heterostructure allowed a selective control of photocatalytic production of H_2_ or CH_4_ from an H_2_O/CH_3_OH mixture. La-doping in all cases increased the photocatalytic activity of SrTiO_3_ nanocatalysts, with the 0.9La:STO showing a benchmark H_2_-production rate of 12 mmol g^−1^ h^−1^. The incorporation of CuO drastically shifted the selectivity from H_2_ toward CH_4_. The highest production was achieved with in tandem incorporation of La and CuO, i.e., La:SrTiO_3_/CuO catalyst, with a CH_4_ production rate of 1.5 mmol g^−1^ h^−1^.

Thus, the present work exemplifies FSP as a potent technology for the production of complex nanocatalysts, at the same time bringing new insights into photocatalysis for H_2_/CH_4_ production.

## Figures and Tables

**Figure 2 nanomaterials-13-00482-f002:**
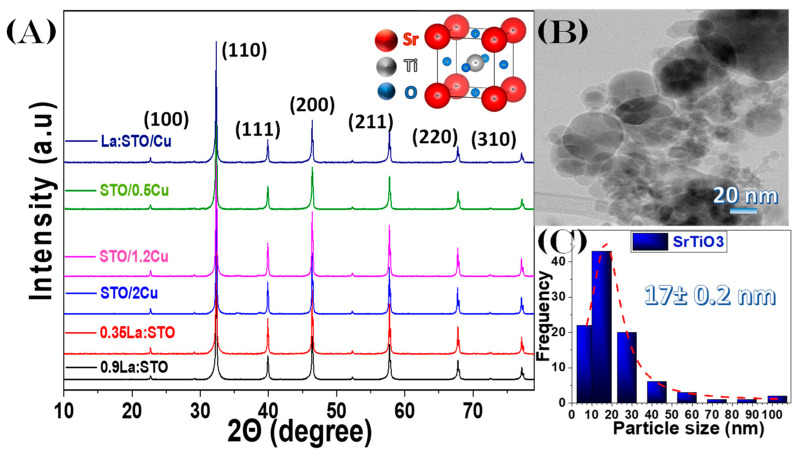
(**A**) XRD patterns for the SrTiO_3_ nanomaterials. *Inset*: The unit-cell structure of the cubic SrTiO_3_ structure. (**B**) TEM image of the pristine SrTiO_3_ particle. (**C**) Size distribution graph obtained from several TEM images.

**Figure 3 nanomaterials-13-00482-f003:**
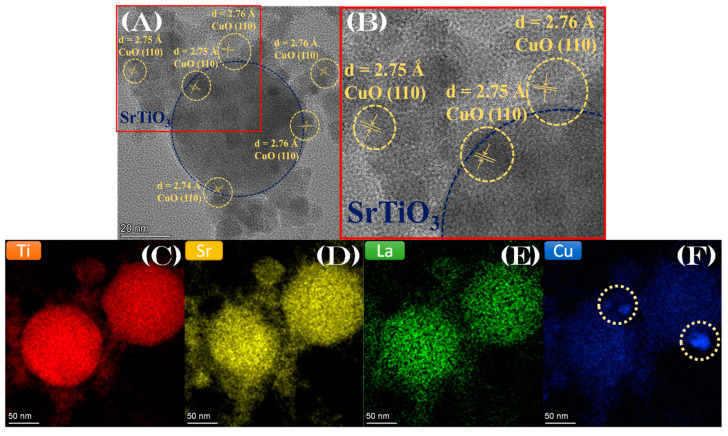
(**A,B**) HRTEM images for the particle La:STO/Cu, with the distinct miller planes of CuO (110) deposited on the SrTiO_3_ surface. (**C**–**F**) Scanning-TEM images with Ti, Sr, La, and Cu atom mapping for material La:STO/Cu.

**Figure 4 nanomaterials-13-00482-f004:**
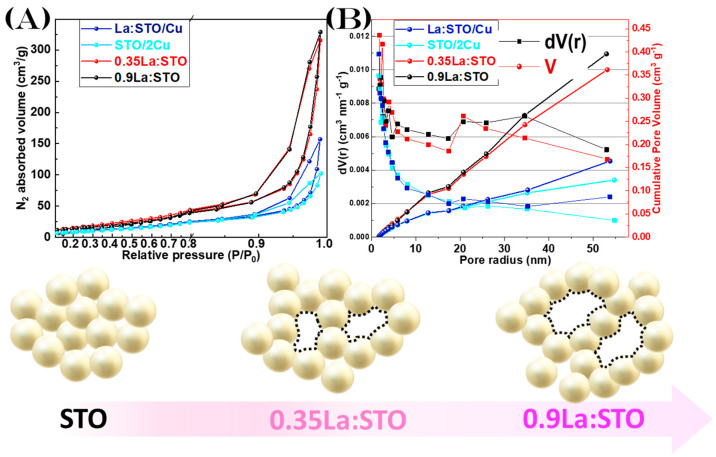
(**A**) Nitrogen adsorption isotherms for the SrTiO_3_-based nanoparticles. (**B**) The corresponding pore-size distribution plot using the BJH method and the Cumulative Pore Volume. *Bottom*: Schematic visualization of the La-doping effect on the nanomaterial aggregation.

**Figure 5 nanomaterials-13-00482-f005:**
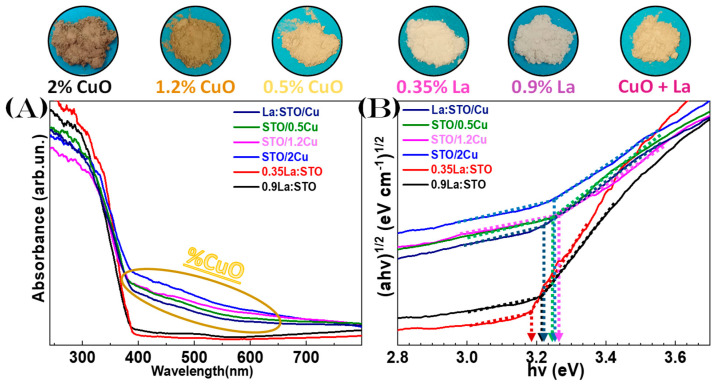
(**A**) DRS-UV-Vis data for the SrTiO_3_ particles. (**B**) Tauc-Plots with the dotted arrows show the calculated bandgap. *Side-photos:* The powder nanoparticles showcase the color shift.

**Figure 6 nanomaterials-13-00482-f006:**
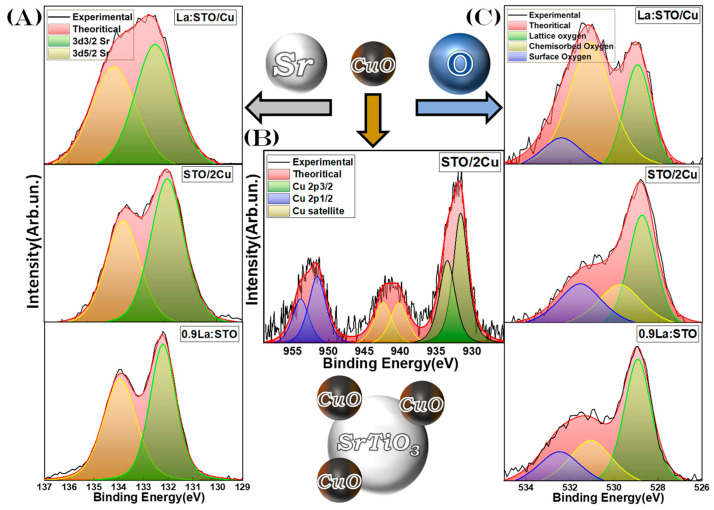
(**A**) Sr 3d_5/2_ and Sr 3d_3/2_ XPS spectra for 0.9La:STO, STO/2Cu, and La:STO/Cu materials. (**B**) Cu 2p_3/2_ and Cu 2p_1/2_ XPS spectra of STO/2Cu for the binding energies. (**C**) Oxygen XPS spectra of 0.9La:STO, STO/2Cu, and La:STO/Cu.

**Figure 7 nanomaterials-13-00482-f007:**
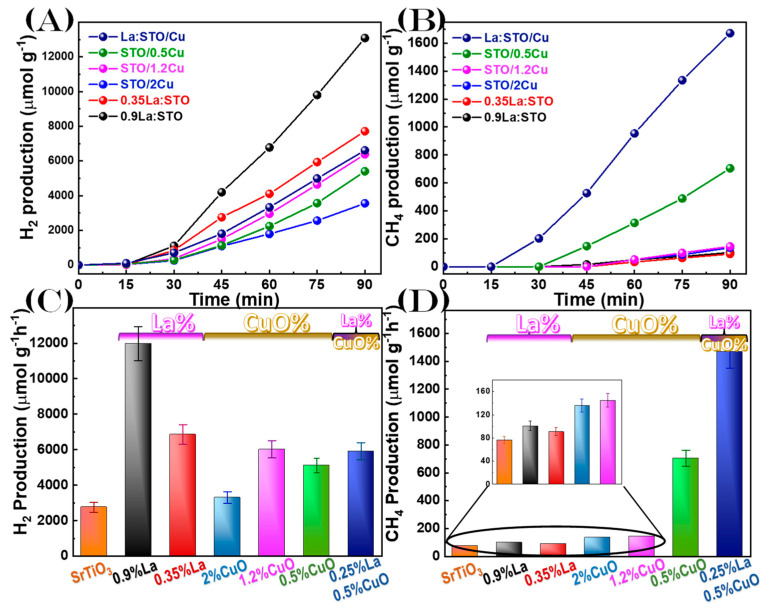
Photocatalytic production by the nanomaterials from an H_2_O/CH_3_OH mixture. (**A**) H_2_-gas vs. time. (**B**) CH_4_-gas vs. time. (**C**,**D**) the corresponding gas production rates (umol per gr catalyst per hour).

**Table 2 nanomaterials-13-00482-t002:** Hydrogen production rates and conditions reported for similar particles/methods.

Photo Catalyst	Synthesis Method	Cocatalyst (wt%)	Light Source	Reaction Solution	H_2_ Yield (umol g^−1^ h^−1^)	Ref.
0.9%La: SrTiO_3_	FSP	1% Pt	Hg (250 W)	H_2_O + 20% CH_3_OH	11,978	This work
0.25%La:SrTiO/0.5% CuO	FSP	1% Pt	Hg (250 W)	H_2_O + 20% CH_3_OH	5907	This work
Ag-STO	Microwave-assisted hydrothermal	2% Ag	UV-low pressure Hg	50% H_2_O + 50% ethanol	463	[63]
Pt- SrTiO_3_	Polymerized-Complex	0.32% Pt	Hg (500 W)	Pure water + 40% methanol	3200	[64]
Nanofibers SrTiO_3_	Electrospinning method	-	Hg (450 W)	Pure water + 40% methanol	160	[65]
La: CoO/SrTiO_3_	Solid statereaction	-	Hg (400 W)	Pure water + Na_2_CO_3_	2800	[66]
Na:SrTiO_3_	polymerizable complex	0.3% Rh	Hg (450 W)	Pure Water	1600	[67]
Macroporous SrTiO_3_	emulsion polymerization method	0.7% Pt	Xe (300 W)	Pure water + 25% methanol	3599	[68]
g-C_3_N_4_/Rh-SrTiO_3_/RGO	Multiple methods	0.58% Rh	Xe (260 W)	Pure water + 10% TEOA	3467	[69]

## Data Availability

Not applicable.

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
