# Peer review of "In Tandem Control of La-Doping and CuO-Heterojunction on SrTiO3 Perovskite by Double-Nozzle Flame Spray Pyrolysis: Selective H2 vs. CH4 Photocatalytic Production from H2O/CH3OH"

_nanomaterials, 2023, doi:10.3390/nano13030482_

Round 1

Reviewer 1 Report

In this manuscript the authors developed high crystallinity SrTiO3 NPs with La-doping in tandem of SrTiO3/CuO heterojunction via a Double-Nozzle Flame Spray pyrolysis method. La:SrTiO3 without CuO produced significantly higher H2, while La:SrTiO3 with CuO can improve production of CH4 selectively. The physical and chemical properties were characterized by XRD, XRF, XPS, BET, UV-vis and DRS. The results can support their hypothesis. However, there are some considerations need to be addressed.

1.      Please correct the pressure number on line 206, which ‘-9’ need to be superscript.

2.      There is lack of statistical significance study for figure 6. In addition, there is lack of error bar for the figure 6. I would suggest conduct duplicate or triplicate of the H2 and CH4 production experiment in Figure 6.

3.      I also recommend the authors have their manuscripts checked by an English language native speaker before final approval of their submission.

Reviewer 2 Report

This manuscript presents a novel La-Doping and CuO-Heterojunction on SrTiO3 Perovskite-based material for photocatalytic production from H2O/CH3OH and is of interest as a potential alternative route. Some points need to be addressed before publication.

 Comments:

1.     The authors need to shorten the introduction section.

2.     Authors need to provide the TEM images of controlled materials.

3.     Authors need to include the limitation of material in the manuscript.

4.     Check the English throughout the manuscript.

Round 2

Reviewer 2 Report

I recommend accepting the work in its current form without further changes.